# END-TO-END LEARNABLE MASKS WITH DIFFERENTIABLE INDEXING

**Dibyanshu Shekhar**[*]   **Sree Harsha Nelaturu**[*]   **Ashwath Shetty**[*]   **Ilia Sucholutsky**
Texas A&M University   Saarland University   Saarland University   Princeton University

## ABSTRACT

An essential step towards developing efficient learning algorithms involves being able to work with as little data as possible to achieve good performance. For this reason, sparse representation learning is a crucial avenue of computer vision research. However, sparsity-inducing methods like importance sampling rely on non-differentiable operators like masking or top-K selection. While several tricks have been proposed for getting gradients to flow 'through' the pixels selected by the operators, the actual indices for which pixels are masked or selected are non-differentiable and thus cannot be learned end-to-end. We propose and implement three methods for making operations like masking and top-k selection fully differentiable by allowing gradients to flow through the operator indices, and show how they can be optimized end-to-end using backpropagation. All three methods can be used as simple layers or submodules in existing neural network libraries.

## 1 INTRODUCTION

Masking and top-K selection are important tools in machine learning for functions like routing (in which various aspects of the downstream task are passed on to a mixture of experts (Masoudnia & Ebrahimpour, 2014)), sparsifying neural networks by dropping out certain connections (Molchanov et al., 2017), and attention-based coordinate evaluations in images or other high dimensional data such as in Correia et al. (2019), Zhao et al. (2019). In short, when trying to mask out certain elements of a tensor based on the values of the same (or another) tensor, top-k is a commonly used approach. Unfortunately, the top-k operator is a discontinuous function and is generally not differentiable. Existing work has used tricks like the Gumbel-max (Jang et al., 2016) or the straight-through method (Bengio et al., 2013) and training with fixed masks Sung et al. (2021) to enable gradients to pass 'through' the mask; however, none of these methods have focused on getting gradients to flow to the actual indices associated with the masked values. In this paper, we focus on the problem of creating end-to-end learnable masks by making the masking indices differentiable. We propose three methods and compare their performance on a simple validation task.

## 2 END-TO-END MASKING WITH DIFFERENTIABLE INDICES

**Method 1: Gumbel Approximation** This method takes advantage of the Gumbel-max trick (Jang et al., 2016) by approximating a deterministic function to evaluate a mask by generating soft labels. The network corresponding to this method takes in a $H \times W$ mask into an encoder block consisting of 2 convolutions followed by a decoder block comprised of 2 deconvolution layers. The final representation from the above blocks is passed through the Gumbel-max operators applied across all channels, the results of which are added together to get the final mask. Depending on the use case, a sigmoid, scaling factor, or clipping can be applied to the final mask.

**Method 2: Gaussian Approximation** This approach is influenced by the challenges of working in the discrete image setting, where the masking operation is non-differentiable, making it unclear how to assign gradients to the assignment operator $M[x, y] = 1$. To overcome this issue, we introduce a novel approach where a one-pixel mask $M_1$ with $M_1[x, y] = 1$ at a specific pixel location $(x, y)$ is represented as a 2D Gaussian with mean $(x, y)$ and covariance $1/delta * I$, as shown in 1. This

---

[*]Equal contribution. Correspondence to: `dshekhar@tamu.edu`

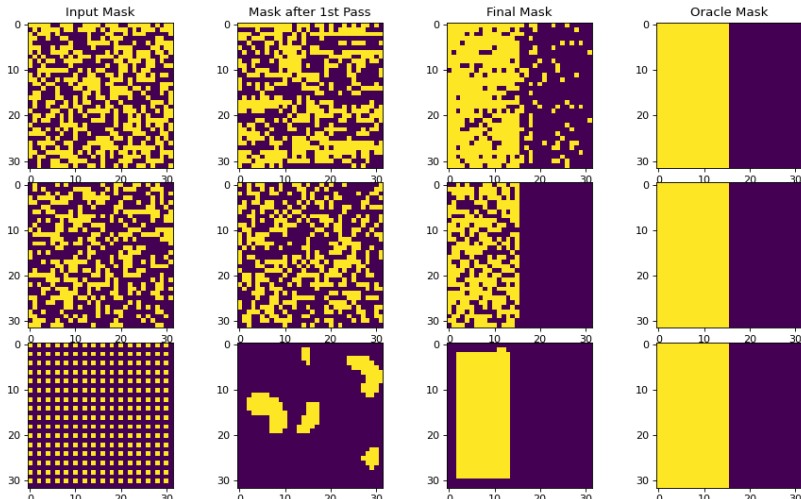

Figure 1: Visualizing each method being used with gradient descent to recover the target mask. Each row contains the input, first step, last step, and oracle mask for each method. **Top**: Gumbel approximation. **Middle**: Gaussian approximation. **Bottom**: Learned model-based operator.

representation enables us to assign gradients to the masking operation in a continuous manner. Using this representation, a new mask $M_2$ with $M_2(a, b) = 1$ can be represented as a 2D Gaussian with mean $(a, b)$ and covariance $=1/\delta * I$, and $\delta$ is the pixel width. The transition from one mask to another can then be seen as a movement of the means of these continuous Gaussians, and this will generate valid gradients since the Gaussian probability function is differentiable.

**Method 3: Learned Model-Based Mask Operator** Neural networks have been used as a faster approximation of numerical operators such as in (Chen et al., 2017). Suppose we have a matrix $V \in \mathcal{R}^{K \times 2}$, where K is the number of active pixels. Each entry in this matrix corresponds to a non-zero pixel location. Let us consider a simple binary mask $M \in \{0, 1\}^{H \times W}$ where $H$ and $W$ are the height and width of the mask. We define a neural network that maps the pixel coordinates to a binary mask such that $f(M; \theta) : R^{K \times 2} - > \{0, 1\}^{H \times W}$. In practice, the output space of the neural network is a continuous relaxation of the binary mask such that $f(M; \theta) : R^{K \times 2} - > [0, 1]^{H \times W}$. We experiment with an MLP-based method to learn this operator using gradient descent. A reconstruction loss between the oracle mask and the one predicted by the model. Once the model converges, the weights are frozen, and it is now a soft approximation of the masking operator.

## 3 RESULTS AND CONCLUSION

We evaluate all three methods on a simple task where an oracle sees a secret target mask and a model-predicted mask and returns only the MSE between the two masks. The goal is to reconstruct the secret mask without actually seeing it, by backpropagating the loss produced by the oracle. We visualize the results for one such oracle mask in Fig 1. These results confirm that gradients indeed flow through the masking indices, and that these gradients can be used to learn masks in an end-to-end way. Future work is needed to determine how these methods scale to different downstream tasks in order to make a principled recommendation about which of the three methods should be used for which applications. The code for the experiments carried out in this paper is available on Github.[1]

---

[1]https://github.com/the-vmlr-lab/Restrictive_Sampling/tree/main/code_diff_indexing

URM STATEMENT

The authors acknowledge that at least one key author of this work meets the URM criteria of ICLR 2023 Tiny Papers Track.

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
