# OpenReview forum: "End-to-End Learnable Masks With Differentiable Indexing"
_ICLR.cc/2023/TinyPapers — Submitted to Tiny Papers @ ICLR 2023_

### Official Review · Reviewer_ofNf · 2023-04-01

**Confidence:** 3

**Summary Of Contributions:**

In this paper, the authors focus on the problem of differentiability of sparsity inducing methods. To this end, this paper proposes three methods for making sparsity-inducing operation fully differentiable. In turn, this allows the methods to be used as layers and submodules within neural networks.

**Rating:**

High Potential (HP): a submission which meets the reviewing criteria and has potential to make an impact on the field

**Strengths And Weaknesses:**

**Strengths**
* S1: The three methods have been explained clearly and concisely
* S2: Experiments have been presented to evaluate the three methods.
* S3: clear directions have been identified for future work

**Weaknesses**
* W1: Hyperparameter values and other details used for experimenting are not shared which can be a challenge if someone id trying to reproduce the work.

**Suggested Changes:**

C1: Some graphics can be added in the appendix, explaining each method visually. This would allow the readers to understand the three methods in a better way.

---

### Official Review · Reviewer_Evv2 · 2023-04-02

**Confidence:** 4

**Summary Of Contributions:**

This paper focuses on the problem of creating end-to-end learnable masks by making the masking indices differentiable. The authors study three different methods and compare their performance on a simple validation task.

**Rating:**

Clear, Correct, and Reproducible (CCR): a submission which meets the reviewing criteria

**Strengths And Weaknesses:**

**Clarity and Correctness**

The paper is well-written and easy to follow. Method discussion is clear and concise. While the authors evaluate the three methods on a simple (single) validation task, there is no discussion of the results (for example, comparing the three methods among themselves; the relation between the final predicted mask in Fig. 1, and the method itself, etc). This makes it hard to derive key takeaways from the work.



**Suggested Changes:**

Overall, I do like the work and strongly believe it's a relevant problem to tackle. I would love to increase the rating of the paper, if (at least some of) the following changes can be incorporated:
- I urge the authors to open-source the code after the review period. Also, I also strongly believe that the readers would significantly benefit from a pseudocode implementation of the three proposed methods (perhaps in the appendix).
- At its current stage, the authors only present results on predicting a single target mask. While I appreciate the simplicity of the evaluation protocol, it's too little to derive any conclusions. For example, are certain masks easier to learn by certain methods; I also observe certain patterns in the final predicted mask, is there any relation to the method itself; etc.

---

### Official Review · Reviewer_7eiM · 2023-04-03

**Confidence:** 5

**Summary Of Contributions:**

The authors propose to solve the problem of learning sparse masks end-to-end during training. They do this by training an MLP model to optimize reconstruction loss measured against an oracle.

**Rating:**

Needs Clarification (NC): a submission which does not meet the reviewing criteria and needs clarification for its described problem or solution

**Strengths And Weaknesses:**

The problem statement is valid. The need for differentiabiity is a known aspect of generating sparse masks.

The work seems very behind on literature survey. Learning sparse masks has had some attention and there are a variety of techniques under investigation that employ this idea of learning the masks The authors have the right sense of the problem and should formulate it correctly in the empirical setting (see a few references below).

The experiment is quite preliminary and the evaluation is missing.

Gonc ̧ alo M. Correia, Vlad Niculae, and Andr  ́e F. T. Martins. Adaptively sparse transformers. CoRR,
abs/1909.00015, 2019. URL http://arxiv.org/abs/1909.00015.

Guangxiang Zhao, Junyang Lin, Zhiyuan Zhang, Xuancheng Ren, Qi Su, and Xu Sun. Explicit sparse transformer: Concentrated attention through explicit selection, 2019.
Sung, Y.-L., Nair, V., and Raffel, C. Training neural networks with fixed sparse masks. In Beygelzimer, A., Dauphin, Y., Liang, P., and Vaughan, J. W. (eds.), Advances in Neural Information Processing Systems, 2021

**Suggested Changes:**

Method 3 is where the paper shold begin even in its correct form. Methods 1 and 2 use terms that are not defined until later.

The evaluation cannot be limited to the visualization. The authors say future work is needed to determine how these methods scale to
different downstream tasks - that is where this work needs to go.

---

### Author Response · Authors · 2023-05-31
**Opt-in**

We wish to opt-in for archival.

---

### Comment · Area_Chair_9TWf · 2023-06-06
**Archival**

This work meets the threshold for archival, contains the URM statement (without specifying which author satisfies the requirements) and is deanonymized.

---

### Meta-Review · Area_Chair_9TWf · 2023-04-04

**Recommendation:** Invite to archive
**Confidence:** 4

**Metareview:**

- Clarity: the paper is overall clear, presenting the problem and methods concisely and straight to the point. The relevant literature is only barely mentioned, making it difficult to position the paper in a wider context
- Correctness: the three proposed methods seem correct, while the lack of literature and experiments makes it difficult to assess the impact of the proposed analysis. No quantitative evaluations are provided.
- Reproducibility: the methods are simple, and some details to reproduce the experimental setting are provided. Code release is not mentioned.

Overall, the reviewers agree on the problem's relevance, and all manage to understand the key passages of the paper. The limited evaluation and literature review harm the potential impact of the paper, which seems more like a positioning one rather than a preliminary experimental study.

**Summary:**

The paper proposes three methods to learn masks in a differentiable way. It poses an interesting problem, pointing to a possible solution, but provides limited evaluation and literature review; also, it does not provide a focused takeaway.

**Comments And Feedback To The Authors:**

The paper is indeed straightforward, and the reviewers appreciated the problem formulation. Improving the literature review, starting from the suggestions given by Reviewer 7eiM is an important step to improve the paper's impact. Adding a proper evaluation would also be important. It might highlight further important aspects of the proposed methods: the obtained masks have three different behaviour, which would be nice to investigate further. A good option is to review similar works and see what the relevant applicative scenarios are. Even considering toy settings (e.g., MNIST, CiFAR-10) would be essential to elaborate further on the paper's findings, clarifying the positioning and the future directions.

**Reason For Not Giving A Higher Recommendation:**

The paper experiments are too limited to derive any conclusion. Also, the little literature review does not provide the reader with a clear context for the work. As a position paper with a single qualitative experiment, I consider the missing literature an important shortcoming.

**Reason For Not Giving A Lower Recommendation:**

The paper is overall clear, points to an interesting problem, and proposes three methods. While the experiment is limited to a single visualization, the shown pattern might rise interesting discussion in the ICLR community.

---

### Decision · Program_Chairs · 2023-04-07

Invite to archive